# Liver Injury and Use of Contrast-Enhanced Ultrasound for Evaluating Intrahepatic Recurrence in a Case of TACE-Refractory Hepatocellular Carcinoma Receiving Atezolizumab-Bevacizumab Combination Therapy: A Case Report

**DOI:** 10.3390/diagnostics11081394

**Published:** 2021-08-01

**Authors:** Satoshi Komiyama, Kazushi Numata, Katsuaki Ogushi, Makoto Chuma, Reiko Tanaka, Sawako Chiba, Masako Otani, Yoshiaki Inayama, Masayuki Nakano, Shin Maeda

**Affiliations:** 1Chemotherapy Department, Yokohama City University Medical Center, 4-57 Urafune-cho, Minami-ku, Yokohama 232-0024, Kanagawa, Japan; skomiyam@yokohama-cu.ac.jp; 2Gastroenterological Center, Yokohama City University Medical Center, 4-57 Urafune-cho, Minami-ku, Yokohama 232-0024, Kanagawa, Japan; k_ogushi@yokohama-cu.ac.jp (K.O.); chuma@yokohama-cu.ac.jp (M.C.); 3Division of Diagnostic Pathology, Yokohama City University Medical Center, 4-57 Urafune-cho, Minami-ku, Yokohama 232-0024, Kanagawa, Japan; g51093@yokohama-cu.ac.jp (R.T.); motani@yokohama-cu.ac.jp (M.O.); inayama@yokohama-cu.ac.jp (Y.I.); 4Department of Clinical Laboratory, Yokohama Medical Center, National Hospital Organization, 3-60-2 Harajuku, Totsuka-ku, Yokohama 245-8575, Kanagawa, Japan; chiba.sawako.bf@mail.hosp.go.jp; 5Tokyo Central Pathology Laboratory, 838-1, Utsukimachi, Hachioji-shi, Tokyo 192-0024, Japan; masayukinakano23@gmail.com; 6Division of Gastroenterology, Yokohama City University Graduate School of Medicine, Yokohama 236-0004, Kanagawa, Japan; smaeda@med.yokohama-cu.ac.jp

**Keywords:** hepatocellular carcinoma, atezolizumab, bevacizumab, liver injury, contrast-enhanced ultrasound

## Abstract

A 67-year-old male with type 2 diabetes (T2DM) was diagnosed with postoperative intrahepatic recurrence for hepatocellular carcinoma (HCC). Nine sessions of transarterial chemoembolization (TACE) proved ineffective, and the patient was diagnosed as having TACE-refractory disease and received seven cycles of atezolizumab–bevacizumab combination therapy. After that, the patient developed hyperglycemia with the HbA1c elevation and the marked fasting serum C-peptide reduction and was diagnosed with developed immune-mediated diabetes (IMD) (T2DM exacerbation with insulin-dependent diabetes development). Subsequently, the hepatobiliary enzyme levels, which were high before the systemic therapy, worsened. Thus, we clinically diagnosed an exacerbation of liver injury due to TACE-induced liver injury complicated by drug-induced liver injury such as immune-mediated hepatotoxicity (IMH). Meanwhile, after contrast-enhanced computed tomography revealed complete response, contrast-enhanced ultrasound was performed to assess intrahepatic recurrence. We found that the latter modality allowed earlier and more definitive diagnosis of intrahepatic recurrence of HCC. Subsequently, despite systemic therapy discontinuation and steroids administration, the liver injury worsened, and the patient died. The autopsy revealed intrahepatic recurrence of HCC and extensive arterial obstruction by the beads used for TACE within the liver, which indicated that disturbed circulation was the primary cause of the liver injury and histopathologically confirmed IMD, but not IMH.

## 1. Introduction

According to world statistics in 2020, liver cancer is the third leading cause of death from cancer and is the 6th most commonly diagnosed cancer in the world. According to the same statistics, hepatocellular carcinoma (HCC) accounts for 90% of all cases of liver cancer [1,2]. Treatment for HCC is often selected according to the disease stage assessed by the Barcelona Clinic Liver Cancer (BCLC) HCC staging system. Presence of multiple intrahepatic nodules in a Child–Pugh class A or B patient with an Eastern Cooperative Oncology Group performance status (ECOG-PS) of 0 is classified as BCLC stage B disease, if distant metastasis and portal vein involvement are absent. For such cases, transarterial chemoembolization (TACE) is the recommended treatment according to existing guidelines [3,4,5,6]. For TACE-refractory BCLC stage B HCC patients, on the other hand, systemic therapy is recommended [4,5,6].

In regard to systemic therapy for HCC, treatment with tyrosine kinase inhibitors, such as sorafenib and lenvatinib, which target the tumor vascular endothelial growth factor (VEGF), have been demonstrated to be effective. On the other hand, monotherapy with nivolumab or pembrolizumab, which are immune-checkpoint inhibitors (ICIs) that inhibit the programmed cell death 1/programmed death ligand 1 (PD-1/PD-L1) system, has failed to show efficacy as first-line systemic therapy [7,8,9,10,11]. PD-L1/PD-1 inhibitors exert their antitumor activity primarily through T cell priming and activation by tumor antigen presentation and destruction of tumor cells by cytotoxic T cells within the tumor microenvironment [12,13]. Furthermore, anti-VEGF drugs are known to reinforce the antitumor activity of ICIs through T cell priming and activation by stimulation of dendrocyte maturation, promotion of cytotoxic T-lymphocyte invasion of the tumor following tumor vessel normalization, and stimulation of reprogramming into immune-permissive tumor microenvironment by suppressed proliferation of myeloid-derived suppressor cells and regulatory T cells [13,14,15,16,17]. It was expected that use of atezolizumab (an anti-PD-L1 antibody) in combination with bevacizumab (an anti-VEGF antibody) might provide synergistic antitumor effect, and this was confirmed in the GO30140 study (a phase Ib trial) and IMbrave 150 study (a phase III trial) [18,19]. In the IMbrave 150 study, atezolizumab–bevacizumab combination therapy significantly extended the survival of patients with unresectable HCC as compared to sorafenib therapy and was, therefore, positioned as first-line standard systemic therapy for unresectable HCC [20].

In regard to immune-related adverse events (irAE), which are adverse events specific to ICIs, multiple guidelines recommend treatments tailored to the irAE type and the severity grades of the AEs rated according to the Criteria Terminology for Adverse Events (CTCAE). However, because of the low incidence of irAEs, the clinicopathological characteristics have yet to be fully clarified, and no prospective large clinical studies of the proposed therapeutic strategies have been carried out yet [21,22,23].

We recently administered atezolizumab–bevacizumab combination therapy to a patient with TACE-refractory HCC, who then showed exacerbation of the liver injury due to multiple factors and developed immune-mediated diabetes (IMD). We performed contrast-enhanced ultrasound (CEUS) using perfluorobutane microbubbles (Sonazoid^®^, Daiichi Sankyo, Tokyo, Japan) in addition to contrast-enhanced CT (CECT) for the detection of intrahepatic recurrence. Furthermore, we carried out an autopsy of the patient and histopathological evaluation of irAE and HCC recurrence. To the best of our knowledge, this is the first case report of autopsy performed on a patient with HCC treated with atezolizumab–bevacizumab combination therapy.

## 2. Case Presentation

### 2.1. Clinical Course

The patient was a 67-year-old Japanese male. At the age of 64, the patient had undergone pancreatoduodenectomy and choledochojejunostomy to treat extrahepatic bile duct cancer. At that time, the extrahepatic bile duct cancer was rated as pT4N1M0, pStage IVA, according to the Union for International Cancer Control (UICC) 7th edition. The patient also had underlying type 2 diabetes mellitus disease, for which he had been prescribed therapeutic diet modification and habitual exercise. Tests for hepatitis B virus and hepatitis C virus markers were negative. There was no family history of endocrine disease (including diabetes mellitus), autoimmune disease, liver disease, or malignancy. The patient was a habitual alcohol drinker, but daily alcohol consumption amount was less than 30 g. During management at a preceding medical facility, he had undergone S4/8 subsegmental hepatectomy to treat HCC, with resection of a single confluent multinodular type of tumor (measuring 55 mm in maximum diameter) that was histopathologically rated as pT2N0M0, pStage II, according to the UICC 7th edition. Six months after the operation, the patient was detected to have developed intrahepatic recurrence and was treated by 6 sessions of drug-eluting bead TACE (DEB-TACE) at the same previous facility. No systemic therapy (including adjuvant chemotherapy) was administered. Forty-one months after the subsegmental hepatectomy, the patient visited our hospital, wishing to continue treatment for recurrent HCC. The first CECT performed at our hospital revealed that the recurrent tumor was confined to the liver (three lesions, 37 mm in maximum diameter), without any evidence of major vessel involvement or distant metastasis, which was within up-to-seven criteria [5]. Therefore, the patient received DEB-TACE with cisplatin, conventional TACE (cTACE) with miriplatin, and cTACE treatment with epirubicin (one session of each), but all proved ineffective. Since no disease control of the recurrent HCC was achieved even after nine sessions of TACE, we judged that the patient had TACE-refractory HCC [24].

We selected systemic therapy as the following line of treatment and registered the patient into the GO30140 study [18]. At the time of registration, the patient was classified as Child–Pugh class A and modified albumin–bilirubin (ALBI) Grade 1, his ECOG-PS was 0, and blood tests revealed elevation of the serum aspartate aminotransferase (AST) (50U/L), alanine aminotransferase (ALT) (59U/L), and alkaline phosphatase (ALP) (518U/L) levels. We considered TACE as the major cause of the liver injury detected in the patient. However, as the body mass index was high (30.2) and the patient had underlying type 2 diabetes mellitus, we considered that he might also suffer from nonalcoholic fatty liver disease (NAFLD) [25]. The serum alpha-fetoprotein (AFP) and protein induced by vitamin K absence-II (PIVKA-II) levels were both within normal range, while the serum HbA1c value was slightly elevated (6.9%). Table 1 shows the blood test data at the start of treatment within the framework of the study that the patient was enrolled in. CECT performed at the start of treatment revealed intrahepatic recurrence in S1, S2, and S8, without any major vessel involvement, lymph node metastasis, or distant metastasis. Fifty-five months after the earlier subsegmentectomy, the patient was started on atezolizumab–bevacizumab combination therapy (atezolizumab 1200 mg/body, bevacizumab 15 mg/kg, once every 3 weeks). The patient developed no adverse events including fever in the first 4 weeks following treatment. A CECT repeated at 15 weeks after the start of the therapy revealed no lesion in S1 or S2, a decrease in the diameter of the lesion in S8, and there were no hypervascular enhanced lesions within the liver in the arterial phase. We judged that the patient achieved partial response according to Response Evaluation Criteria in Solid Tumors version 1.1 (RECIST v1.1) and complete response according to the modified Response Evaluation Criteria in Solid Tumors (mRECIST). Figure 1 graphically represents the changes over time in the imaging findings of the lesions in S8 and S1 (Figure 1a–c: CECT before the start of treatment, Figure 1d,e: CT at 25 weeks after the start of treatment, Figure 1f: CT scan at 28 weeks).

Figure 2 illustrates the time course of the hepatobiliary enzyme levels and clinical course of the patient after the start of atezolizumab–bevacizumab combination therapy. We continued the atezolizumab–bevacizumab combination therapy, and seven cycles had been administered by 18 weeks after the start of the therapy. At 21 weeks after the start of the therapy, the patient complained of poor appetite and malaise. The blood test revealed hyperglycemia (569 mg/dL), elevation of the serum level of beta-hydroxybutyric acid (658 μmol/L), and slight elevation of the plasma osmotic pressure (295 mOSm/kg), accompanied by further elevation of the serum HbA1c level to 8.9%. Furthermore, the fasting serum C-peptide (1.11 ng/mL) was higher than the lower limit of the normal range, and the serum test for glutamate decarboxylase antibodies was negative. Acidemia was absent in a venous blood sample and urinalysis revealed a negative result for urinary ketones (qualitatively). Thus, the patient’s condition, not satisfying the criteria for the diagnosis of diabetic ketoacidosis, was judged as reflecting exacerbation of type 2 diabetes mellitus [26]. The possibility of hypothyroidism was ruled out on the basis of the blood test data for free thyroxine (T4) and thyroid-stimulating hormone, and the possibility of hypopituitarism was ruled out on the basis of the blood test data for adrenocorticotropic hormone, luteinizing hormone, follicle-stimulating hormone, and cortisol. However, a repeat blood test on the following day revealed a marked reduction in the fasting serum C-peptide level (0.4 ng/mL). Considering this finding with the elevated serum level of beta-hydroxybutyric acid, we considered the possibility that the patient had developed insulin-dependent diabetes, possibly corresponding to IMD caused by atezolizumab. At that time, we discontinued the atezolizumab–bevacizumab combination therapy and initiated treatment with rapidly acting insulin analogs and a long-acting insulin analog. In addition, treatment with empagliflozin was started on day 10 after the start of insulin therapy, and the HbA1c level improved to 8.2%.

Twenty-one weeks after the atezolizumab–bevacizumab combination therapy was started, the blood test revealed CTCAE v5.0 Grade 1 elevation of AST, ALT, and bilirubin, as well as Grade 2 elevation of ALP. No evidence of dyslipidemia was noted, and the IgG levels did not increase. Tests for both antinuclear antibody and antimitochondrial antibody were negative. CECT revealed no stenosis, dilatation, or wall thickening of the bile duct. On the basis of these findings, the patient was clinically diagnosed with exacerbation of liver injury due to TACE-induced liver injury complicated by drug-induced liver injury such as immune-mediated hepatotoxicity, and the atezolizumab–bevacizumab combination therapy was discontinued because these two drugs were suspected as the culprits responsible for the drug-induced liver injury complicating the TACE-induced liver injury. After discontinuation of these drugs, the ALP rose even further, and the patient was started on treatment with methylprednisolone for treatment to immune-mediated hepatotoxicity. Thereafter, the methylprednisolone dose level was reduced steadily, while confirming the improvement of the laboratory data. The patient was continued on maintenance steroid therapy (15 mg prednisolone/day) from the 12th day of treatment. In the blood test performed at 25 weeks after the start of treatment, Grade 3 elevation of AST, ALT, and ALP and Grade 1 elevation of bilirubin were noted. At that time, the patient was started on ursodeoxycholic acid; however, this failed to have any marked beneficial effect. The oral prednisolone dose was increased to 40 mg/day, which resulted in alleviation of the liver injury, allowing us to steadily reduce the prednisolone dose. On the basis of the findings of CEUS at 26 weeks after the start of treatment, we diagnosed intrahepatic recurrence of HCC. However, we postponed resumption of the atezolizumab–bevacizumab combination therapy in view of the possibility that this therapy was responsible for the exacerbation of liver injury in the patient. Twenty-eight weeks after the start of treatment, the blood test revealed elevation of the ALP and bilirubin. At that time, we repeated a CT to explore the cause of these changes in the hepatobiliary enzyme levels and found an abscess in S8 of the liver. Percutaneous drainage of the liver abscess was performed, along with initiation of antibiotic treatment, while the prednisolone treatment was continued. The liver abscess diminished in size temporarily, but liver injury worsened again. CT performed at 32 weeks after the start of treatment revealed numerous new abscesses in the liver. The patient’s general condition was poor, and we judged that the patient was not a suitable candidate for any invasive procedure. Therefore, the patient was only given antibiotic therapy. The patient eventually died 33 weeks after the start of treatment.

### 2.2. Findings of Diagnostic Imaging upon Intrahepatic Recurrence of HCC

In the non-enhanced abdominal CT performed at 25 weeks after the start of atezolizumab–bevacizumab combination therapy, the HCC lesion in S8, which had been found at the start of the therapy, was visualized as a low-density lesion and did not show any hyperenhancement in the arterial phase of CECT. Laboratory examination at that time revealed elevation of the serum level of PIVKA-II, and we performed CEUS at 26 weeks (Figure 3). B-mode imaging showed a tumorous lesion measuring 20 mm in diameter in S8, and we judged that this lesion was identical to the S8 lesion detected on the CT performed at the start of treatment. When perfluorobutane microbubbles were used for contrast enhancement, this lesion was visualized as a hypervascular enhanced mass in the arterial phase, as a perfusion defect in the post-vascular phase, and again as a hypervascular enhanced mass following reinjection of perfluorobutane microbubbles, on the basis of which we diagnosed intrahepatic recurrence of HCC [27]. Later, in the arterial phase of the CECT performed at 28 weeks after the start of treatment, this lesion assumed the form of a slightly enhanced mass, so that we diagnosed intrahepatic recurrence of HCC.

### 2.3. Autopsy Findings

When the patient died, autopsy was performed with consent from the family. Histopathologically, the liver showed the features of interlobular cholangitis and related abscess formation, but there were no evidence of immune-mediated hepatotoxicity and no plasma cell infiltration or interface hepatitis as histopathological signs of autoimmune hepatitis. Meanwhile, extensive arterial obstruction by the beads used for DEB-TACE was noted, surrounded by massive/submassive necrosis. Green-colored nodular lesions (25 mm and 20 mm, respectively) were noted in S1 and S8 of the liver. Both lesions showed signs of coagulative necrosis in most parts, but their periphery contained atypical cells with swollen nuclei (with evident nucleoli) and silver staining revealed fine- to medium-sized trabecular structures. Immunohistochemically, the atypical cells showed positive staining for glutamine synthetase, negative staining for glypican 3, and positive staining for CD34 along the sinusoidal capillaries. On the basis of these findings, the nodular lesions in S1 and S8 were diagnosed as moderately differentiated HCC (Figure 4).

Histopathological examination of the pancreas revealed atrophy of the pancreas, fibrous hyperplasia between the acini or around the duct, and mild lymphocytic infiltration, and based on the findings, we made the diagnosis of chronic pancreatitis. Immunohistochemically, positive staining for anti-synaptophysin antibodies showed islets, but not anti-insulin antibodies, which led us to conclude that the beta cells had disappeared. Meanwhile, the islets showed positive staining for anti-glucagon antibodies, and we concluded that the alpha cells remained (Figure 5).

## 3. Discussion

In the present patient with TACE-refractory HCC, we started treatment with atezolizumab–bevacizumab combination therapy, which led to the patient showing worsening of the underlying liver function and developing features of IMD (new onset of insulin-dependent diabetes accompanied by worsening of the underlying type 2 diabetes mellitus). For the assessment of intrahepatic recurrence, we performed CEUS with perfluorobutane microbubbles in this case, which enabled us to make an earlier and more definitive diagnosis of intrahepatic recurrence of HCC as compared to CECT. At autopsy, we histopathologically confirmed the liver injury caused by repeated TACE, IMD, and intrahepatic recurrence of HCC.

Liver injury has been reported to occur at an incidence rate of 4–14% following cTACE and 30–36% following DEB-TACE [28,29]. In the IMbrave 150 study, the incidence of liver injury was 43.2% in the atezolizumab–bevacizumab group, comparable with the sorafenib group (39.7%). The majority of these events were laboratory abnormalities (atezolizumab–bevacizumab group: 38.3%; sorafenib group: 34.6%). The clinical diagnosis of liver injury was 13.1% in the atezolizumab–bevacizumab group and 12.8% in sorafenib group, of which the most commonly reported event was ascites (atezolizumab–bevacizumab group: 7.0%; sorafenib group: 5.8%) [19]. Additionally, the incidence of liver injury after ICI therapy has been reported to be higher in patients with HCC than in patients with other types of cancer [30]. This difference is probably attributable to one or more of the following factors: progression and relapse of intrahepatic tumor lesions and a high prevalence of underlying chronic liver diseases, such as viral hepatitis, alcohol-related liver disease, NAFLD, and autoimmune hepatitis [21,30]. Atezolizumab–bevacizumab combination therapy after TACE resulted in exacerbation of the liver injury in our patient. These events seemed to be probably attributable to: (1) disturbed circulation through the liver caused by the beads used for DEB-TACE, (2) liver injury caused by atezolizumab, (3) bevacizumab-induced inhibition of angiogenesis. In relation to liver injury caused by atezolizumab, it has been reported that in HCC patients with underlying non-alcoholic steatohepatitis (NASH), the liver tissue contains an abundance of CD8^+^PD-1^+^ T cells, which may be involved in liver tissue damage [31]. Although the histopathological findings in our patient did not satisfy the criteria for the diagnosis of NASH, the clinical findings suggested the presence of underlying NAFLD, so that we assumed that the liver injury in our patient also was possibly attributable to liver tissue damage caused by atezolizumab via the mediation of CD8^+^PD-1^+^ T cells.

One known mechanism for the onset of IMD is ICI-induced inhibition of the function of PD-L1 expressed on islets. IMD often shows clinicopathological features similar to those of type 1 diabetes mellitus, as they both show absolute insulin deficiency due to pancreatic beta cell destruction by the autoimmune system [32,33]. Concretely, it is not uncommon for patients with IMD to reduce the fasting serum C-peptide levels to below the detection limit and develop complicating diabetic ketoacidosis [32,34]. In our patient reported herein, the clinical findings suggested sudden insulin deficiency, and autopsy confirmed marked reduction in the beta cells alone in the islets, leading us to diagnose insulin-dependent diabetes arising from absolute insulin deficiency [35]. We additionally diagnosed exacerbation of type 2 diabetes mellitus by the clinical findings. While a patient with IMD often shows a disease type akin to type 1 diabetes mellitus (acute autoimmune insulin-dependent diabetes), there are reported cases of IMD that show relative insulin deficiency as exacerbation of type 2 diabetes mellitus (type 2 diabetes phenotype). Marchand et al. reported that the clinical manifestations differ depending on the presence or absence of a history of type 2 diabetes mellitus, in that the former often do not develop complicating diabetic ketoacidosis and the serum C-peptide levels remain measurable [36,37]. Therefore, we may consider the present patient as a case of IMD in which acute autoimmune insulin-dependent diabetes and the type 2 diabetes phenotype coexisted.

In regard to evaluation of the responses to treatment in clinical studies of treatment of malignant neoplasms, the RECIST (a set of criteria based on changes of the tumor diameter compared to pre-treatment assessment) is often used. For evaluation of the responses of HCC treatment, the mRECIST is also frequently used, in which changes in the tumor blood flow are added to changes in the tumor diameter [38,39,40]. In a published study designed to evaluate the responses of HCC to treatment with angiogenesis inhibitors, it was reported that the prognosis was reflected better according to the mRECIST than the RECIST, indicating the usefulness of tumor blood flow evaluation [41]. Though CT and MRI are recommended as initial imaging modalities for assessing response to treatment in the RECIST and the mRECIST, there are reports of the higher sensitivity of CEUS than that of CECT for detecting residual HCC after TACE [38,39,42,43]. Higher detection of the residual HCC by CEUS could be attributed to several reasons: one can be the difference in sensitivity to hypervascularity. CEUS with perfluorobutane microbubbles has been reported in the literature as being more sensitive than CECT for detecting hypervascularity in both advanced and early cases of HCC [44,45]. When CECT is used, there are cases in which the imaging cannot be performed at the appropriate timing for evaluation of a hypervascular enhanced mass in the arterial phase. CEUS, on the other hand, can yield images of the same cross-section chronologically, thereby allowing evaluation at the appropriate timing, even when the timing of the contrast material inflow deviates slightly from the optimal timing, which leads to the adequate evaluation of the tumor vascularity [44]. Additionally, in the present case, CEUS allowed earlier and more definitive detection of intrahepatic recurrence of HCC than CECT. Thus, CEUS is also promising for evaluation of the responses of patients with HCC to systemic therapy.

In the present case, the patient’s tumor was judged as TACE-refractory on the basis of the Japan Society of Hepatology Liver Cancer Study Group of Japan (JSH-LCSGJ) criteria 2014 [24,46]. The effectiveness of systemic therapy has been reported for patients with TACE-refractory HCC [5]. Thus, we selected atezolizumab–bevacizumab combination therapy as a subsequent therapy. The systemic therapy available for advanced HCC cases includes sorafenib, lenvatinib, regorafenib, ramucirumab besides atezolizumab–bevacizumab combination therapy. Each of these agents has been recommended for cases rated as Child–Pugh class A [3,4,5,6,20]. It has also been reported that among the patients of unresectable HCC treated with lenvatinib, cases rated as ALBI Grade 1 showed a higher response rate and a lower percentage of cases requiring discontinuation of the drug owing to the emergence of adverse events [47]. Additionally, concerning immunotherapy, Pinato et al. reported that the median overall survival of unresectable HCC patients treated with ICIs were significantly stratified by ALBI grade (grade 1, 22.5 months; grade 2, 9.6 months; grades 3, 4.6 months; *p* < 0.001) [48]. Therefore, liver function is essential both as a determinant of indication and a prognostic factor in patients with advanced HCC treated with systemic therapy. In patients with BCLC stage B HCC, Hiraoka et al. reported that ALBI-score significantly worsened in association with repeated TACE procedures [49]. Thus, to avoid deterioration of liver function, it is essential to judge TACE refractoriness adequately and switch to systemic therapy as soon as possible.

## 4. Conclusions

Our experience with this HCC case treated by repeated TACE and atezolizumab–bevacizumab combination therapy yielded three important findings and one important suggestion: (1) multiple factors may be involved in the development/progression of liver injury; (2) IMD can assume a form other than insulin-dependent diabetes; (3) evaluation of tumor vascularity by CEUS may be useful for the diagnosis of intrahepatic recurrence. Our experience provides an important suggestion that an early switch from repeated TACE to systemic therapy may be useful to avoid unnecessary deterioration of liver function in patients with TACE-refractory HCC.

## Figures and Tables

**Figure 1 diagnostics-11-01394-f001:**
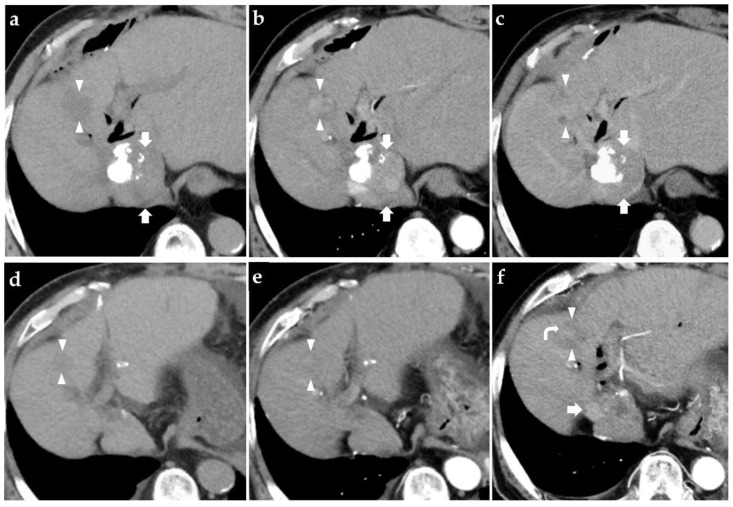
Changes over time in the hepatocellular carcinoma (HCC) lesions in S8 and S1. (**a**) A non-enhanced abdominal CT performed before the start of treatment revealed a low-density area in the S1 (arrow) and S8 (arrowhead) of the liver. (**b**,**c**) On contrast-enhanced CT performed prior to the start of treatment, both the lesions in S1 (arrow) and S8 (arrowhead) are visualized as hypervascular enhanced lesions in the arterial phase and washout in the equilibrium phase, which led to the diagnosis of viable HCC lesions. Similar findings were also noted in S2, and these lesions were also judged as being viable HCC lesions (data not shown). (**d**) A non-enhanced abdominal CT obtained at 25 weeks after the start of treatment revealed a low-density area in S8 (arrowhead). (**e**) On contrast-enhanced CT obtained at 25 weeks after the start of treatment, in the arterial phase, no hypervascular enhanced mass was visualized in either S8 (arrowhead) or S1 (data not shown). (**f**) In the arterial phase of contrast-enhanced CT performed at 28 weeks after the start of treatment, a part of the lesion in S8 (arrowhead) is visible as a faintly increased attenuation (curved arrow). The arrow indicates the inferior vena cava.

**Figure 2 diagnostics-11-01394-f002:**
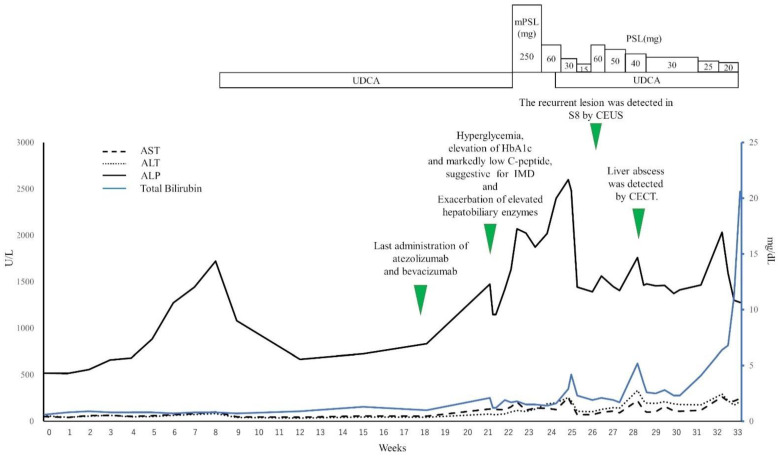
Clinical course and changes over time of the hepatobiliary enzyme data after the start of atezolizumab–bevacizumab combination therapy. At 21 weeks after the start of atezolizumab–bevacizumab combination therapy, the patient developed hyperglycemia, accompanied by further elevation of the HbA1c level as compared to the pretreatment level. The fasting serum C-peptide level decreased within a short period of time, followed by exacerbation of liver injury. CEUS with perfluorobutane microbubbles performed at 26 weeks after the start of treatment revealed hypervascularity area in S8 during the arterial phase, as well as a defective area during the post-vascular phase, suggestive of intrahepatic recurrence of hepatocellular carcinoma. CECT at 28 weeks revealed a liver abscess. AST, aspartate aminotransferase; ALP, alkaline phosphatase; ALT, alanine aminotransferase; CECT, contrast-enhanced CT; CEUS, contrast-enhanced ultrasound; IMD, immune-mediated diabetes; mPSL, methylprednisolone; PSL, prednisolone; UDCA, ursodeoxycholic acid.

**Figure 3 diagnostics-11-01394-f003:**
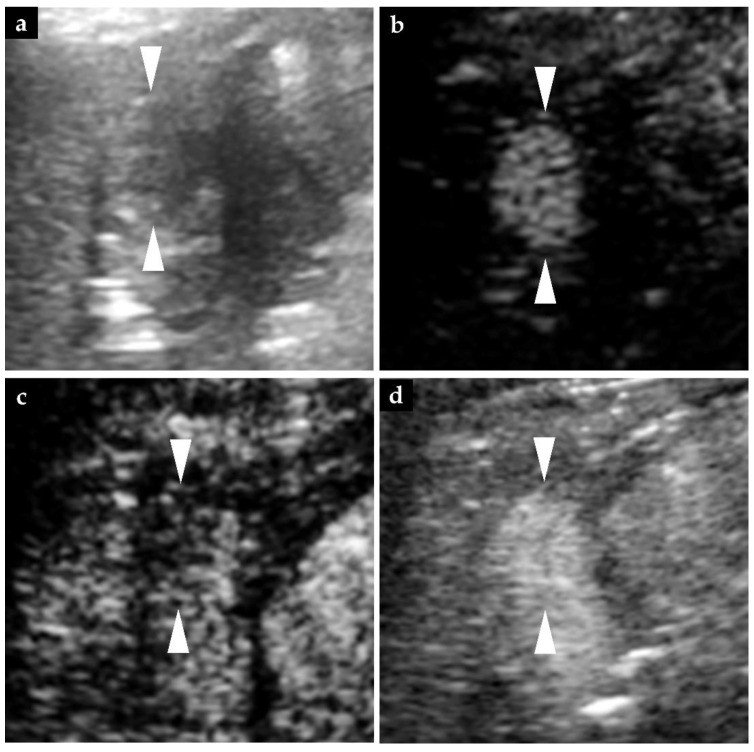
Ultrasound findings of the HCC lesion in S8 at 26 weeks after the start of treatment. (**a**) B-mode image showing a 20 mm HCC lesion in S8. (**b**,**c**) On contrast-enhanced ultrasound with perfluorobutane microbubbles, a hypervascular enhanced mass is visible in S8 during the arterial phase obtained 21 s after injection, and a defective area is noted during the post-vascular phase obtained 10 min after injection. (**d**) Twenty-four s after re-injection during the post-vascular phase after the initial injection of perfluorobutane microbubbles resulted in a greater hypervascular enhancement of the lesion in S8 as compared to the surrounding area, confirming that the lesion was a hypervascular tumor. Arrowheads indicated the margin of the lesion.

**Figure 4 diagnostics-11-01394-f004:**
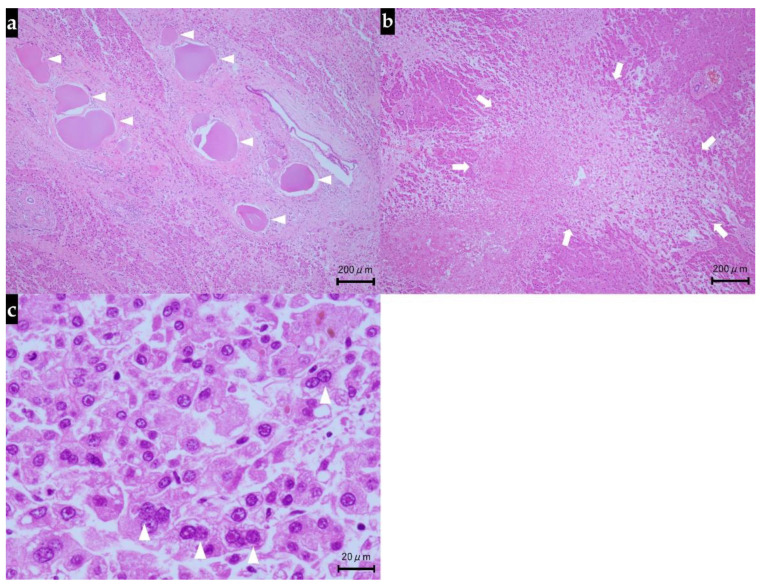
Histological findings of the liver at autopsy (hematoxylin and eosin staining). (**a**) Arterial obstruction by DEB-TACE beads were visible in the portal area (arrowheads). (**b**) Extensive necrosis was seen around the central vein (arrows). (**c**) Obvious cancer cells with deformed nuclei (arrowheads) including hypercellularity and increased nuclear–cytoplasmic ratio were observed as the features of moderately differentiated hepatocellular carcinoma; DEB-TACE, drug-eluting bead transarterial chemoembolization.

**Figure 5 diagnostics-11-01394-f005:**
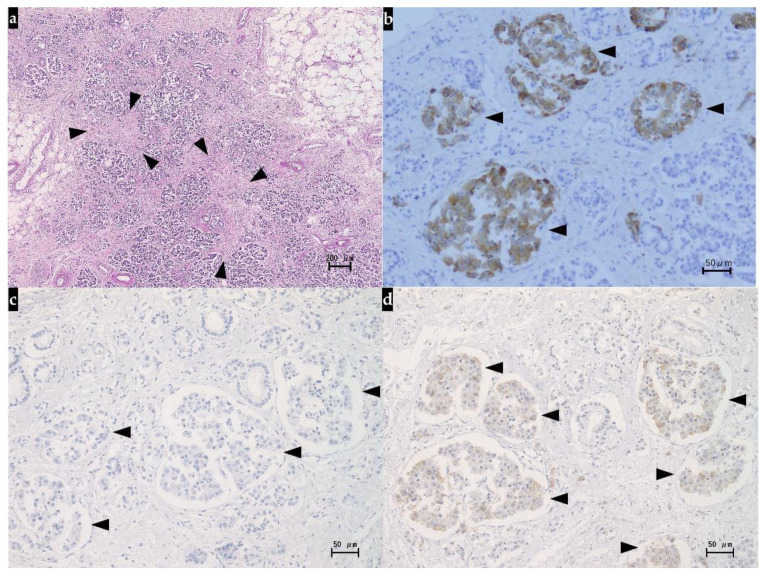
Histological findings of the pancreas at autopsy. (**a**) Hematoxylin and eosin staining showed intense fibrosis (arrowheads) and mild inflammatory cell infiltration among the acini, accompanied by marked acinar atrophy. (**b**) Islets were identified by positive immunohistochemical staining with anti-synaptophysin antibodies (arrowheads). (**c**) None of the islet cells showed positive immunohistochemical staining with anti-insulin antibodies, indicative of the loss of beta cells (arrowheads). (**d**) Islets showing positive immunohistochemical staining with anti-glucagon antibodies were noted sporadically, suggesting the presence of residual alpha cells (arrowheads).

**Table 1 diagnostics-11-01394-t001:** Laboratory test results at the start of the atezolizumab–bevacizumab combination therapy.

	Values	Normal Range
Peripheral blood		
White blood cell count (/μL)	4910	3300–9400
Red blood cell count (10^4^/μL)	532	414–534
Hemoglobin (g/dL)	15.1	13.8–17.2
Platelet count (10^4^/μL)	11.0	18.0–39.0
Prothrombin time (INR)	1.02	0.90–1.10
Blood chemistry		
Sodium (mmol/L)	140	138–144
Potassium (mmol/L)	4.4	3.7–5.0
Chloride (mmol/L)	105	100–108
Calcium (mg/dL)	9.2	8.8–10.1
Total protein (g/dL)	6.7	6.9–8.3
Albumin (g/dL)	4.1	4.2–5.4
Total bilirubin (mg/dL)	0.6	0.4–1.8
Aspartate aminotransferase (U/L)	50	14–32
Alanine aminotransferase (U/L)	59	11–45
Alkaline phosphatase (U/L)	518	109–312
Gamma-glutamyl transpeptidase (U/L)	351	10–58
Lactate dehydrogenase (U/L)	243	116–199
Amylase (U/L)	242	43–130
Total cholesterol (mg/dL)	156	<219
Triglyceride (mg/dL)	124	<149
Blood urea nitrogen (mg/dL)	13	8–20
Creatinine (mg/dL)	0.84	0.68–1.04
Glucose (mg/dL)	114	<110
HbA1c (%)	6.9	4.6–6.2
Immunological tests		
C-reactive protein (mg/dL)	0.401	<0.20
Hepatitis B surface antigen/antibody	both negative	
Hepatitis B core antigen/antibody	both negative	
Hepatitis C virus antibody	negative	
Tumor markers		
Alpha-fetoprotein (ng/mL)	6	<10
Protein induced by vitamin K absence or antagonists-II (mAU/mL)	27	<40
Carcinoembryonic antigen (ng/mL)	3.4	<5.0

## Data Availability

Not applicable.

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
