# Peer review of "Liver Injury and Use of Contrast-Enhanced Ultrasound for Evaluating Intrahepatic Recurrence in a Case of TACE-Refractory Hepatocellular Carcinoma Receiving Atezolizumab-Bevacizumab Combination Therapy: A Case Report"

_diagnostics, 2021, doi:10.3390/diagnostics11081394_

Round 1
Reviewer 1 Report
This is a very interesting and well performed case report evaluating the effect of atezolizumab-bevacizumab combination therapy in a patient with TACE-refaractoru HCC and the role of CEUS with perfluorobutane microbubbles in the evaluation of HCC intrahepatic recurrence. Atezolizumab-bevacizumab combination therapy after TACE resulted in exacerbation of the liver injury provbably related to mutiple factors: altered hepatic circulation caused by the beads used for DEB-TACE, atezolizumab related live injury, and bevacizumab-induced antiangiogenetic effect.
The authors hightlighted the importance of CEUS in the prediction of intrahepatic HCC recurrence compared to CECT.
The interesting conlcusion of the study was that an early switch from repeated TACE to systemic therapy may be useful to avoid unnecessary deterioration of liver function in patients with TACE-refractory HCC.
The study is well designed and well written with clinical relevant conclusions and appropiate data supporting the results.
Author Response
Response to Reviewer 1 comment:
Thank you for your careful review of our manuscript. We are thankful for the time and energy you expended.
Reviewer 2 Report
An interesting case report. I have some suggestions that may help the authors to improve the paper.
Lines 74-87. I believe this paragraph would be more appropriate for the discussion section.
The value of CT is not clear to me. I would appreciate it if you could explain the value more.
Figure3. Please add the time points in CEUS images. (For example 4 sec after injection etc..)
Figure4. Please annotate figures 4b and 4c. Same comment for figure5.
Thank you for considering my recommendations.
Author Response
Responses to Reviewer 2 Comments:
Thank you for your careful review of our manuscript. We are thankful for the time and energy you expended. We have read the comments carefully and revised the manuscript accordingly. The revised portions in the manuscript have been written in red. We have also responded to the comments in a point-by-point manner as follows.
- Lines 74-87. I believe this paragraph would be more appropriate for the discussion section.
Thank you so much for your valuable comments. According to your comments, we moved the paragraph explaining liver injury in the Discussion section of the revised manuscripts (page 10-11, lines 309-321). With this movement, the position of the defined abbreviation for nonalcoholic fatty liver disease (NAFLD), conventional transarterial chemoembolization (cTACE) and drug-eluting bead transarterial chemoembolization (DEB-TACE) and the citation number have changed (page 3, lines 106, 113 and 124–125 of the revised manuscript).
- The value of CT is not clear to me. I would appreciate it if you could explain the value more.  
Thank you so much for your valuable comments. In general, CT and MRI are recommended as initial imaging modalities for assessing response to treatment both in the RECIST and the modified RECIST. Following your suggestions, to make better understandings of the CT value, we have incorporated the phrase explaining the importance of CT scans in the RECIST and the modified RECIST (page 11, lines 359–361 of the revised manuscript).
- Figure 3. Please add the time points in CEUS images. (For example 4 sec after injection etc..) 
Thank you so much for your valuable comments. At your suggestion, we have added the time points in CEUS images in Figure 3 (page 8, lines 256-257 of the revised manuscript).
- Figure 4. Please annotate figures 4b and 4c. Same comment for figure 5.
Thank you so much for your valuable comments. According to your comments, we have added arrows and arrowheads to indicate the location and extent of the histopathological findings in figures 4b, 4c, and 5a-d. With these modifications, we have revised the sentences explaining the annotations and histopathological findings in the figures (page 9, lines 284-289, page 10, lines 291–298 of the revised manuscript).
Other modifications
According to the author contributions regarding clinical management of the patient, we have added Katsuaki Ogushi and Makoto Chuma as co-authors. While, we have deleted Satoshi Moriya from the co-authors. Therefore, we have revised co-authors (page 1, lines 6 and 11-12 of the revised manuscript). We apologized that we have changed the co-authors.
Round 2
Reviewer 2 Report
Thank you for considering my recommendations. I do not have any further suggestions.